# Food Systems and Planetary Health Nexus Elective: A Novel Approach to a Medical Education Imperative for the 21st Century

Modan Goldman †, Aditya Vaidyam †, Sindhu Parupalli †, Holly Rosencranz, Davendra Ramkumar and Japhia Ramkumar *

Carle Illinois College of Medicine, University of Illinois Urbana-Champaign, 506 S Mathews Ave, Urbana, IL 61801, USA; modanrg2@illinois.edu (M.G.); vaidyam2@illinois.edu (A.V.); sindhup2@illinois.edu (S.P.); harosen@illinois.edu (H.R.); ramkumar@illinois.edu (D.R.)
* Correspondence: jayasing@illinois.edu
† These authors contributed equally to this work.

**Abstract:** This is a report on an inaugural medical student elective, *Microbiomes Matter: The Path to Regenerative Systems of Farm, Food, and Health*, from the perspective of the student participants. Recognizing food as medicine is gaining support across many settings. However, little is known about how medical schools engage in this holistic approach. Integrating food systems and the connections to soil and human health through microbiomes into medical education represents a transformative shift towards more holistic healthcare practices. We describe the course content and impact of a medical school elective in food systems. This elective employed a systems lens and planetary health perspective to explore the impact of climatic factors and environmental degradation on farms, nutrition, and non-communicable lifestyle diseases. Through the two-week course, medical students gained insights into sustainable food systems, supply chains, and the importance of regenerative agriculture. The course also provided a comprehensive overview of the gut microbiome, nutrition, technologies, and the economics of food systems, including their impact on lifestyle diseases. By fostering a systems-oriented mindset, this elective better equips medical students to address the complex challenges of human and planetary health and promote regenerative, sustainable, culturally sensitive, and robust systems of farm, food, and health.

**Keywords:** planetary health; food systems; regenerative agriculture; microbiomes; sustainability; climate change; nutrition; medical education





## 1. Introduction

Climate change and biodiversity loss pose an urgent and interrelated crisis that has gained attention from the global community and demands immediate action. A joint statement by over 200 health journals calls on the United Nations, political leaders, and health professionals to recognize that climate change and biodiversity loss are inseparable challenges [1,2]. They emphasize that addressing these issues together is critical for preserving human health and averting a catastrophic global health emergency. In considering the training of medical professionals, the Clinicians for Planetary Health initiative represents a global endeavor to mobilize health professionals, patients, and communities to address planetary health through lifestyle modifications and activism. This initiative recognizes that healthcare providers can influence positive changes in individual behaviors and advocate for broader societal actions to address the pressing issues of climate change and biodiversity loss. The severity of this global crisis cannot be understated and it underscores the need for a concerted effort to mitigate its impacts.

The 2022 Challenges article by McLean et al. describes a "medical education planetary health journey" and chronicles an encouraging growing momentum in planetary health

education and engagement [3]. This movement is further exemplified by the planetary health report card (PHRC), a student-led initiative that assesses and aims to improve planetary health education in medical schools worldwide [4]. Since its inception in 2019, the PHRC has grown to encompass 15 countries and over 105 medical schools, demonstrating its value as a tool for medical schools to improve their planetary health education and for students to advocate for planetary health within their institutions and communities. While a significant step forward in medical education, there is yet more to be done today to address the urgency and severity of the planetary health crisis. There is wide support for these endeavors in medical education. According to Wellberry et al., "Fortunately, many health topic areas already exist in medical school curricula where climate change education can be incorporated into the discussion simply by broadening the horizon within which these topics are taught" [5]. The incorporation of content supporting the food as medicine approach initiative, developed by the United States (US) Department of Health and Human Services, is an ideal opportunity [6]. Additionally, the Liaison Committee on Medical Education, recognized by the US Department of Education and the World Federation of Medical Education, has standards for the function and structure of a medical school which includes item 7.5, "Societal Problems: The faculty of a medical school ensure that the medical curriculum includes instruction in the diagnosis, prevention, appropriate reporting, and treatment of the medical consequences of common societal problems" [7].

Medical students (authors MG, AV, and SP) participated in such a curriculum in the summer of 2023. This was in the form of a dedicated two-week non-clinical elective at our institution, Carle Illinois College of Medicine. Such an elective is designed to allow students to pursue scholarly experiences that do not involve direct patient care but that are relevant to clinical medicine. Examples include content that addresses interprofessional teamwork, core science knowledge, population health, and preventative medicine. This elective explored food systems and the connections to soil and human health through microbiomes. It also examined the complex web of factors influencing the quality of our food. This perspective aligns with the principles of regenerative systems thinking, emphasizing the need for sustainable food systems to enhance healthcare outcomes. The novelty of this elective was the incorporation of non-traditional topics that underscore the complex interplay among extractive agricultural practices, unhealthy food systems, and the misuse of antibiotics. These all contribute to the degradation of soil and microbiomes, subsequently diminishing food quality and exacerbating pathological conditions within the human microbiome [8]. Soil health deterioration, intensified by unsustainable systems and the impacts of climate change, further diminishes the nutritional value of plants, ultimately resulting in reduced food quality. These interconnected environmental challenges collectively have a detrimental impact on the human microbiome, highlighting the intricate relationship between soil, the environment, and the human gut [9].

Current agricultural practices, commonly referred to as industrial agriculture (IA) used in the production of food, are a known contributor to climate change largely by the increase in greenhouse gasses [10]. These systems of farming include the use of synthetic fertilizers, herbicides, pesticides, intensive irrigation, genetically modified disease-resistant and high-yielding plants and animals, large-scale monocropping, intensive tillage of the land with decreased or absent fallow periods, and high stocking density in animal husbandry with associated concentrated animal feeding [11]. Climate change can also adversely affect soil processes including its physicochemical properties and the soil microbiome by direct and indirect means [12]. In recent years, there has been increasing interest in the use of regenerative agriculture (RA) in an attempt to break this cycle [13]. RA prioritizes soil health and involves practices such as crop rotation, the use of organic fertilizers, eliminating/minimizing synthetic chemical inputs in crop farming and animal husbandry (e.g., antibiotics and hormones), the use of cover crops, recycling farm waste (composting), integrating livestock into crop production systems, animal welfare, and minimal to no tilling [13,14]. Evidence is emerging in support of the theory that food produced by RA (plant or animal/animal products used for food) is of better quality than foods produced

by IA [15–17]. Examples included increased levels of certain vitamins, minerals, micronutrients, antioxidants, and lower levels of pesticide residue. Some of these benefits may be mediated through the effect of RA on the soil/plant microbiome [18].

Noncommunicable chronic diseases (NCDs) are chronic diseases not directly caused by infectious pathogens or injury nor by maternal or perinatal conditions. They have been linked to lifestyle factors including 'unhealthy' diets [19]. An example is a Western diet, which is characterized by "high intakes of pre-packaged foods, refined grains, red meat, processed meat, high-sugar drinks, candy, sweets, fried foods, conventionally raised animal products, high-fat dairy products, and high-fructose products" [20]. The World Health Organization (WHO) notes that an unhealthy diet is one of the leading risks for the global burden of diseases, mainly those that are noncommunicable [21]. NCDs include cardiovascular diseases (atherosclerotic heart disease and stroke), cancer, metabolic diseases such as diabetes mellitus, obesity, nonalcoholic fatty liver diseases, chronic respiratory disease, various neurologic and psychiatric diseases, inflammatory conditions such as asthma, allergies, and inflammatory bowel disease [19]. In these conditions, dysbiosis, a change in the gut microbiome that is deleterious to the homeostatic interplay with the human host, plays a role [22]. With unhealthy diets, there are microorganisms that predominate, such as *Bacteroides* species, and reduced species diversity (Figure 1) [23].

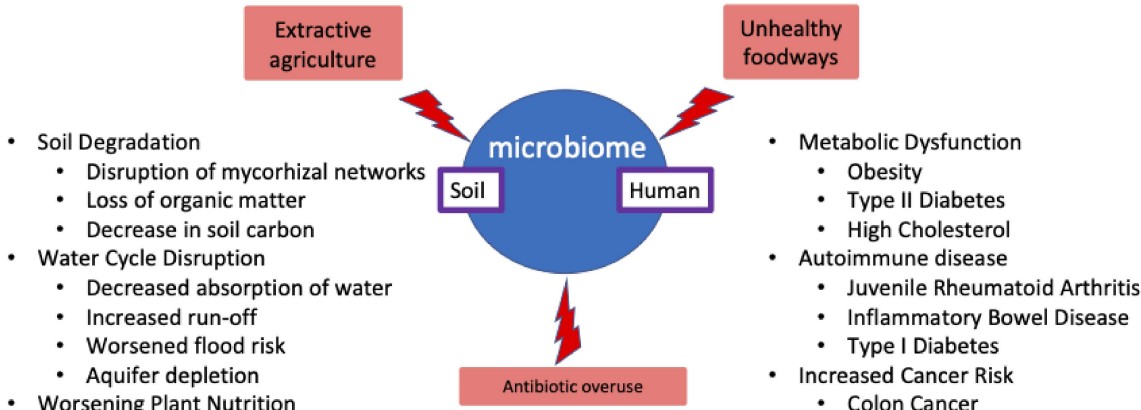

**Figure 1.** The interplay between extractive agriculture, unhealthy food systems, and antibiotic overuse directly impacts soil and human microbiomes.

Dietary manipulation is well established as a modality of therapy in the primary and secondary prevention of NCDs [24]. There is emerging evidence that these benefits are in part derived from the effect of a healthy diet on the gut microbiome [25]. In contrast to the diet that promotes NCDs, a healthy diet is mainly plant-based and rich in vegetables, fruit, legumes, fish, poultry, and whole grains and minimizes red meat, processed foods, and excessive simple sugars. The gut microbiome in healthy individuals consuming this type of diet is notable for the predominance of certain genera such as *Prevotella* and a marked increase in diversity compared with dysbiotic microbiomes [26]. In addition to the traditional metrics used to analyze the nutritional content of food, there is increasing interest in evaluating the role of other macromolecules such as phytosterols in assessing food quality, as these may have direct benefits to human metabolism and may be indirectly of importance via their effect on the gut microbiome [25,27].

As healthcare and medicine continue to undergo this current paradigm shift towards preventative health and holistic well-being, the concept of "food as medicine" emerges as a critical component of this transformation. As medical professionals explore the intricate relationship between nutrition, gut microbiomes, and inflammation, the agricultural sector has embarked on a journey into regenerative agriculture, focusing on soil biodiversity, nutrient exchange, and crop nutrient density [28]. It is also important to understand the economic and policy factors that are involved in order to make changes for the betterment of human and planetary health. Having the skills to advocate for reasonable change from the

standpoint of healthcare providers is also key to this paradigm shift. According to Abbasi et al., "Health professionals must be powerful advocates for both restoring biodiversity and tackling climate change for the good of health" [2]. For example, healthcare professionals can broaden their engagement by advocating for legislation such as the US Farm Bill. This comprehensive legislation aims to mitigate the effects of climate change through funding for conservation programs, sustainable agriculture, and research for farm systems to be more sustainable and resilient. Additionally, there are measures to provide improved access to healthy nutrition to food-insecure populations in order to reduce rates of chronic disease and hunger [29].

There is a gap in medical education where connecting the impact of farm and food systems on human health has been largely overlooked. There is a growing volume of global calls to action for planetary health education in medical curricula [28,30–57]. To that end, this novel and forward-thinking medical elective course was developed which emphasized the importance of understanding the links between regenerative agriculture and human health, encouraging students to trace the origins of many chronic diseases back to food systems and how they are grown, produced, and consumed. It employs a systems lens to explore topics such as the impact of climatic factors on farms and nutrition, NCDs, and the planetary health perspective. NCDs contribute to 74% of global mortality, highlighting the importance of understanding their underlying causes for future physicians [19]. Throughout this elective, medical students were positioned to engage in an immersive experience that took them beyond traditional medical education, offering the opportunity to explore regenerative food systems, defined as the sustainable approach to producing, distributing, and consuming food for the restoration and enhancement of ecological and community health. This course explored how soil health, food quality, and human well-being are intricately intertwined, shaping the health outcomes of individuals and communities alike, as depicted in Figure 2. These factors begin the dialogue towards a deeper understanding of planetary health as well as personalized health in the clinical context at a level suitable for a medical student.

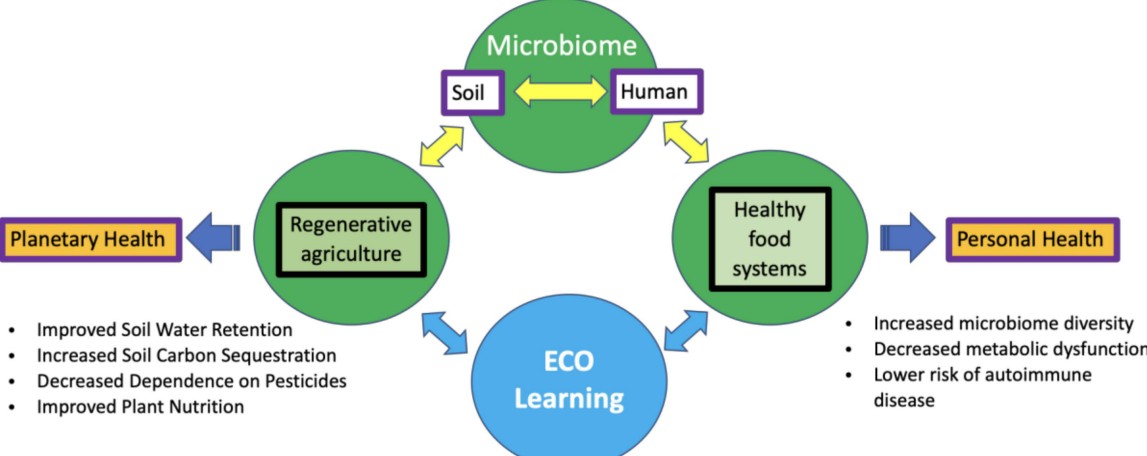

**Figure 2.** The link between the major course objectives: soil and human microbiomes, healthy food systems, and regenerative agriculture.

The elective course embodied a transformative educational experience that bridges regenerative agriculture and human health. It equipped medical students to understand the roots of chronic diseases and empowers them to address these issues from the ground up, embracing the concept of "food as medicine" and viewing food systems as integral to patient care and global health. Through an exploration of this elective course, this paper will provide a preview of the course content, highlighting its relevance to medical students who seek to understand not only the treatment of diseases but also the promotion of health and well-being from a broader planetary perspective. Through a critical reflection and guide

to the implementation of this elective course, the authors hope to bridge the gap between medicine and sustainability, nurturing a new generation of healthcare professionals ready to address the global sustainability challenges of our time.

## 2. Educational Approach

To equip medical students with a planetary perspective on healthcare, this elective course took a multidisciplinary approach. This pedagogy transcended traditional classroom boundaries and included diverse guest lectures, farm and community visits, innovative thought experiments, physician shadowing, and cooking with regeneratively farmed foods for optimal gut health. The ultimate goal of this systems-based method was to examine the interrelationships and interconnectedness of various components within a system, as many real-world issues are multifaceted and cannot be fully understood by looking at elements in isolation. This process encouraged students to view and explore topics holistically, taking into account how different parts of a system interact and influence each other. Combined with the systems-based method, our elective contributes to a comprehensive, innovative, and multifaceted medical curriculum.

A syllabus was developed for this two-week elective by a transdisciplinary team of experts in the fields of medical education, internal medicine, gastroenterology, food systems, nutrition, and agriculture. The syllabus adhered to the requirements of our institution for a nonclinical elective for contact hours, assignments, and assessment. Content experts to deliver educational materials were recruited from the affiliated campus, healthcare system, and community and willingly participated. The educational approach involved three major phases. The initial phase involved traditional classroom-based learning in a synchronous interactive online environment. Components included guest lectures by content experts, thinking exercises (e.g., "Describe 3 takeaways from this course that you can utilize in a clinical setting to counsel patients on healthier, sustainable dietary choices", "What do see as your role in the food system as a change agent"), guided discussions (e.g., "Describe the role and voice of physicians in advocating for systemic change within the hospital system"), and interactive learning (e.g., a visit to a local sustainable farm and visit to the mobile market food bus). Topics covered included merging medical education with agriculture, nutrition, policy, environmental sciences, economics, microbiology, and supply chain analysis. Students were exposed to perspectives from various community stakeholders from organic and regenerative sectors and multiple university disciplines. Students were encouraged to engage in dynamic team-based learning to encourage peer teaching, debate, and discussion (Table 1).

The secondary phase involved experiential learning and clinical education. Students visited local farms to investigate regenerative agricultural practices, mobile markets to explore food justice and community-based solutions to increasing food security, and local hospital food services to explore implementing sustainable food choices into an organization that serves the medical needs of diverse communities. Students shadowed a local gastroenterologist with expertise in the gut microbiome and gained relevant hands-on experience in the clinic.

The final phase involved a culinary medicine session where students had a kitchen space to connect the various systems they learned about in the preparation of a meal. Under the supervision of a registered dietician and chef who was part of the development team, the students gained cooking skills and incorporated new and unfamiliar ingredients culminating in a positive and inspiring shared experience in a meal. Students left the course with an appreciation of where their food comes from, food growing practices, and food-related health impacts.

**Table 1.** The course content outline, providing an overview of lecture material, labeled by presenter(s), topic, and learning environment.

| Learning Environment | Topic | Presenter(s) | Overview |
|---|---|---|---|
| Classroom | Planetary Health | Physician: Clinical Associate Professor Internal Medicine | Introduction of planetary health serves as a foundational lecture for the course. Extractive processes and nature-based solutions help expand students' thought in their approach to understanding problems and applying the process of innovation. |
| | Food Systems | Professor, Food Systems and Nutritional Sciences | Introduction to systems thinking and understanding the role of impact analysis of food choices on sustainability. |
| | Food Transport | Registered Dietician; Non-profit produce distributor | The journey of food, emphasizing the importance of sustainable supply chains and food security. |
| | Food Production | Professor, Agronomy; Director of Illinois Regenerative Agriculture Initiative | Industrial and regenerative agriculture, permaculture, and their societal and clinical implications. Aspects include corn and soybean rotation, miscanthus, and perennial rice, as well as societal and clinical implications of industrial agriculture. |
| | Stakeholders Panel | Farmers processors, distributors, and retailers of sustainable food | Connect students with local stakeholders in the organic and regenerative sectors, providing diverse perspectives from community leaders including farmers, processors, distributors, and retailers. |
| | Gut Microbiome | Physician: Clinical Associate Professor Internal Medicine Gastroenterologist | Overview of the human gut microbiome in health and disease and new areas of research. |
| | Nutrition | Professor, Food System, and Nutritional Sciences | Discussing nutrition in a clinical setting, exploring the role of food habits in patient engagement through interactive platforms such as US MyPlate and Tiny Habits. |
| | Economics | Associate Professor, Sustainable Agriculture and Food Systems; Food systems analyst | Economic aspects of food systems and the costs of lifestyle diseases. |
| | Technologies | Professor, Food System and Nutritional Sciences | Role of existing technology and future innovations in food systems, evaluating its impact on society and cultures. |
| | Advocacy/Policy | Associate Professor and Director, Agricultural Policy Program | The Farm Bill's impact on modern agriculture and food insecurity in the United States. The significance of political activism in medicine. |
| | Food Safety | Associate Professor, Applied Food Safety | Addressing food safety and infectious disease in public health and clinical practice. Understanding biases against aspects of food systems and better reconciling thoughts about industrialization and conventional agricultural practices. |
| | Animal Agriculture | Clinical Veterinarian: Teaching Assistant Professor Public Health and Epidemiology | Understanding sustainable food systems and soil regeneration by evaluating conventional and organic animal husbandry practices. |

**Table 1.** *Cont.*

| Learning Environment | Topic | Presenter(s) | Overview |
|---|---|---|---|
| Clinical | Gastroenterologist Shadowing | Physician: Clinical Associate Professor Internal Medicine Gastroenterologist | Students shadow a gastroenterologist, gaining insights into the clinical context of gut health. |
| Experiential and Community | Mobile Market | Director of hospital community health initiatives | Students visit a community solution for improving food security—a bus that brings free healthy food to underserved populations. |
| | Hospital Food Services | Director of hospital food services | Students explore the application of sustainable food practices in a community hospital system aiming to improve the health of their patients and planet. |
| | Regenerative Farm | Farmer practicing regenerative agriculture | Students visit a local farm practicing regenerative agriculture and community food justice via equitable food distribution. |
| | Culinary Medicine | Chef and Registered Dietician | Preparation of a sustainable and healthy meal to deepen student understanding of food sourcing, food-growing practices, and their relationship with food, nutrition, health, and the planet. |

## 3. Impact on Medical Education

Medical students became active participants in a transformative learning experience. Insights gained from this course offered an understanding of sustainable food systems, supply chains, regenerative agriculture, and the role of microbiomes in human health. They connected the dots between nutrition, economics, culture, and technology, all within the context of food systems. Additionally, the course instilled in them a sense of advocacy, empowering them to take action in promoting community food justice and healthier eating habits.

Students gained a deeper understanding of sustainable food systems and supply chains, recognizing the far-reaching consequences of our food choices. Through this course, they developed an appreciation for regenerative agriculture, both in theory and practice, witnessing its positive outcomes for the environment and society. Furthermore, students now grasped the critical link between the gut microbiome, nutrition, and human health, fostering a holistic perspective on medicine. The course provided them with valuable insights into the economic aspects of food production and how innovation and culture shape our food systems. Students were actively engaged in local initiatives promoting food justice and were empowered to make a difference in their communities through initiatives such as mobile markets, community agriculture, and incorporating sustainable food practices in businesses. In learning more about their own hospital's innovative cafeteria service, medical students experienced a deeply personal connection to the course lecture content, such as sustainability and the importance of local sourcing. They recognized the impact of these ideas on their own 'lunch hour wellbeing' as well as the directly tangible impact of the cafeteria operations on their patients' care and recovery. This experience acted as a lead-by-example anchor for the students, promoting and modeling the behaviors explained throughout the course directly, with direct measurables presented. Additionally, the hands-on culinary medicine experience, promoting healthy eating and living, embodied the elective's mission of creating healthcare professionals with a holistic understanding of planetary health.

The following list was compiled by the students and summarizes their insights gained from the elective course during a debriefing session:

- Practical communication skills:

  ○ Enhanced proficiency in communicating with future patients about dietary choices, emphasizing the crucial link between nutrition and health. To actively promote healthier eating habits in clinical settings, course elements may be integrated into patient education programs, such as hands-on cooking with regeneratively farmed foods;

- Commitment to institutional change:

  ○ Commitment to implementing sustainable food practices within future clinical settings and translating personal choices into a proactive stance on healthcare sustainability. Readiness to influence institutional policies and contribute to broader healthcare sustainability initiatives, to ensure a tangible impact beyond individual actions;

- Cross-disciplinary appreciation:

  ○ Acquired appreciation for cross-disciplinary collaboration in healthcare, recognizing the relevance of non-human fields of study (plant and animal agriculture, economics, and environmental sciences) in shaping holistic healthcare solutions. Eagerness to actively seek collaboration with professionals from diverse fields to integrate a broader perspective into future healthcare initiatives;

- Community health initiatives:

  ○ Pledge to initiate projects, acting on the responsibility towards community health. Plans include collaborating with local farmers, establishing community gardens, and advocating for healthier food options in underserved areas, ensuring a practical contribution to community well-being;

- Educational outreach vision:

  ○ Recognizing the potential for broader impact through educational outreach beyond academic institutions, with the vision of collaborating with community organizations and policymakers. Intent to disseminate knowledge about sustainable food systems actively to reach a broader audience;

- Preventive care approach:

  ○ Utilizing culinary medicine as a proactive form of preventive care with a positive impact on community health, incorporating culinary medicine sessions into community settings, schools, and healthcare institutions. This aims to instill healthy eating habits from an early age, ensuring a practical contribution to long-term community health;

- Political activism for systemic change:

  ○ Commitment to political activism from recognizing the interconnectedness of agriculture, the environment, and healthcare outcomes. This commitment extends to engaging in political advocacy for broader systemic changes, beyond healthcare-exclusive policies.

## 4. Transformative Potential

The integration of food systems and soil health to human health through microbiomes into medical training represents a paradigm shift in healthcare education. A holistic healthcare mindset goes beyond treating symptoms to addressing the root causes of health issues. By fostering advocacy for regenerative and sustainable food systems, future healthcare professionals are equipped with the tools to tackle complex healthcare challenges in a future of planetary crises. Through this elective, healthcare professionals are equipped to contribute to improved gut health, reducing the prevalence of lifestyle diseases, and aligning seamlessly with the principles of planetary health. In this era of planetary crises, this holistic approach to medical education not only prepares students to provide effective

healthcare but also positions them as champions of a healthier and more sustainable future for all.

## 5. Future Directions

Medical schools must collaborate across disciplines and expand their curriculum to include planetary health, systems thinking, and nutrition. Medical educators and policymakers must also educate future healthcare leaders to understand the significance of planetary health as they become future world leaders in policy as well.

The inclusion and expansion of educational opportunities that include food systems and nutrition in medical schools have a pivotal role in healthcare education. It is important that this content be made more widely available and accessible, beyond the reach of a two-week elective that may not impact a large number of students. In line with a commitment to experiential and longitudinal learning, this content should be integrated into the curriculum. Learning about nutrition, the effect of food on the microbiome, and how food is grown and produced embodies a "food as medicine" approach and should be incorporated into applicable didactics and clinical instructional experiences. There are multiple strategies that can be used to amplify and deploy this content such as online modules for asynchronous access during relevant clinical and didactic training and through clinical scenarios presented with standardized patient exercises. Additionally, specific topics can inspire research and service-learning projects where students actively participate in community outreach. To achieve this, there must be interdisciplinary collaboration between medical schools, other academic departments, and community organizations, fostering a rich exchange of knowledge and ideas. The authors recognize the importance of interprofessional education by engaging not only medical students but also sustainably-minded trainees from nursing, pharmacy, veterinary, agriculture, engineering, political science, business, and economics. This collaborative approach nurtures teamwork and a shared commitment to holistic patient care.

There is a responsibility of healthcare professionals to advocate for planetary health and regenerative systems through their practice and policy engagement, aligning with the importance and relevance of initiatives such as climate-smart agricultural practices in the US Farm Bill [28]. Recognizing the essential role doctors play in addressing these matters, our elective actively encourages students to become advocates for positive change. Medical schools have a responsibility to guide future physicians to meet the needs of their patients and community; training in planetary health and the impacts of climate change advocacy can advance this critical role [58]. Furthermore, there is an ongoing need for research and advocacy in this field as medical students emerge as future leaders who contribute to this growing body of knowledge and advocate for sustainable and regenerative food systems. These include more animal and human studies evaluating the relationship between food and the gut microbiome, the links between dysbiosis and NCDs, agricultural methods and the impact on soil and plant microbiomes and thus food quality, and further studies to build the case for regenerative climate-smart agricultural methods. Together, these future directions will equip medical students with the knowledge and skills to make a lasting impact, contributing to planetary health and a sustainable future.

## 6. Conclusions

The participation of medical students in a novel elective on regenerative food systems has been a transformative journey as reflected in their list of insights and serves to close a significant gap in medical education in connecting the impact of farm and food systems on human health. Through the lens of a regenerative systems approach, future healthcare professionals can gain a deeper understanding of the intricate connections between food systems, environmental sustainability, and human well-being. Students learned firsthand the impact of dietary choices on both personal health and the health of our planet. This educational experience instilled a sense of responsibility in them as well as of advocacy, recognizing that they play a vital role in shaping a healthier future for our planet and all its

inhabitants. By embracing the principles of planetary health, students are not only better prepared to address the complex healthcare challenges of the future but also to actively contribute to solutions that promote the well-being of both individuals and the global ecosystem. In a world facing numerous planetary crises, the imperative of aligning medical education with planetary health principles and planetary health has never been clearer. Our novel elective serves to equip the next generation of healthcare professionals with the knowledge, skills, and passion needed to foster regenerative and sustainable systems, ultimately working towards a healthier and more harmonious future for all.

**Author Contributions:** Conceptualization, M.G., A.V., S.P., H.R. and J.R.; writing—original draft preparation, M.G., A.V. and S.P.; writing—review and editing, M.G., A.V., S.P., H.R., D.R. and J.R.; visualization, M.G., A.V., S.P., H.R., D.R. and J.R.; supervision, J.R. and H.R. All authors have read and agreed to the published version of the manuscript.

**Funding:** This elective was made possible by the I-REGEN grant, University of Illinois Urbana-Champaign.

**Data Availability Statement:** No new data were created or analyzed in this study. Data sharing is not applicable to this project report.

**Acknowledgments:** This elective was developed by the Coalition of Regenerative Agriculture Food and Health (CRAFH) team, a collaboration between Basil's Harvest (E. Meyer; K. Bloedorn; and D. Blood), faculty at the University of Illinois Urbana Champaign (J. Ramkumar; D. Ramkumar and H. Rosencranz) and the University of Wisconsin Stevens Point (J. Steinmetz).

**Conflicts of Interest:** The authors declare no conflicts of interest.

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
