# Peer review of "Food Systems and Planetary Health Nexus Elective: A Novel Approach to A Medical Education Imperative for the 21st Century"

_challenges, doi:10.3390/challe15010006_

Round 1

Reviewer 1 Report

Comments and Suggestions for Authors

The report is well written, refers to an actual problem, the interrelationship and interconnection between climate change, biodiversity loss and human health and emphasize the importance to improve planetary health education in medical schools worldwide. In this context, it presents a very interesting and comprehensive outline of the suggested elective course which gives the students the opportunity to deeper understand the link between planetary health and regenerative system through experiential learning and clinical education. Overall, the report appears to be relevant and well-constructed, gives and interesting scientific perspective with potential to make significant contributions to healthcare education.

Author Response

Reviewer 1 Comments and Suggestions

The report is well written, refers to an actual problem, the interrelationship and interconnection between climate change, biodiversity loss and human health and emphasize the importance to improve planetary health education in medical schools worldwide. In this context, it presents a very interesting and comprehensive outline of the suggested elective course which gives the students the opportunity to deeper understand the link between planetary health and regenerative system through experiential learning and clinical education. Overall, the report appears to be relevant and well-constructed, gives and interesting scientific perspective with potential to make significant contributions to healthcare education.

Response

Thank you for your positive feedback and support of our work. We appreciate your support for this significant contribution to the literature.

Reviewer 2 Report

Comments and Suggestions for Authors

Dear authors

We agree on the importance of the topic, given that the planetary health and food systems nexus needs to be introduced in medical and overall health sciences educational curricula, however, details and outcomes of the specific project for a broader audience are poorly explained in the current project report.

Some specific comments are:

- The abstract lacks structure and does not comprehensively inform the purpose and results of the study.

- Although the background of the topic has been well argued, the details and outcomes of a specific project are not included. It is lacking a comprehensive explanation of the methodology, results, and potential implications.

It would have been interesting to have included and shared more practical insights, applications, or lessons derived from the project.

Thus, the details and outcomes of the specific project for a broader audience are poorly explained in the current project report.

Comments on the Quality of English Language

There have been some minor typos, but the overall English language is correct. 

Author Response

Reviewer 2 Comments and Suggestions

Dear authors

We agree on the importance of the topic, given that the planetary health and food systems nexus needs to be introduced in medical and overall health sciences educational curricula, however, details and outcomes of the specific project for a broader audience are poorly explained in the current project report.

Some specific comments are:

- The abstract lacks structure and does not comprehensively inform the purpose and results of the study.

- Although the background of the topic has been well argued, the details and outcomes of a specific project are not included. It is lacking a comprehensive explanation of the methodology, results, and potential implications.

It would have been interesting to have included and shared more practical insights, applications, or lessons derived from the project.

Thus, the details and outcomes of the specific project for a broader audience are poorly explained in the current project report.

Response

Thank you for your comments and suggestions. We have added details and examples to better explain our methodology and outcomes, in addition to adding student insights. We hope these have improved our project report.

Reviewer 3 Report

Comments and Suggestions for Authors

Dear author,

The manuscript title is “A Food Systems and Planetary Health Nexus Elective: A Novel Approach to A Medical Education Imperative for the 21st Century” and it aims to present a pedagogical project developed in a medical school around Planetary Health. The topic falls within the aims and scope of the journal. It is relevant in the context of the urgent need to approach Planetary Health, especially with health students.

Some particular suggestions/comments will be done here:

-      Abstract: reading the abstract and the scope of the manuscript, I miss any reference to animal health as they are also included in food systems and in other zoonoses transmitted in other forms rather than foodborne. One Health is included in Planetary Health and may not be neglected. If the authors want to reach only the vegetable part of food systems/agriculture, I believe Planetary Health is too ambitious for the title

-      Lines 53-54 – the authors should state at least the university/country of this Department

-      Lines 92 – 101 – too many lines for one only reference

-      Line 54 – LCME – first time write in full please

-      Line 104 – Figure 1 – what is “hi” cholesterol?

-      Line 131 – The authors should think they are not writing only to American readers, but for global (planetary!!) ones. So, whenever you refer concrete American projects, you should detail something more about them. This reviewer has no idea about what is “Farm Bill”.

-      Lines 181/182 – why did veterinarians were not included? It is a pity. The “animal” part is still missing and not only in the abstract.

-      Line 202 – ON – how did you assess all this “gain”? Maybe these are the project aims but not necessarily achieved? If you did assess, you should state here how did you do it.

-      Line 238 – One Health and Planetary Health are two fashion areas, but they are two different concepts. Along the manuscript the authors seem to use one or another randomly and that should not happen.

Author Response

Reviewer 3 Comments and Suggestions for Authors

Dear author,

The manuscript title is “A Food Systems and Planetary Health Nexus Elective: A Novel Approach to A Medical Education Imperative for the 21st Century” and it aims to present a pedagogical project developed in a medical school around Planetary Health. The topic falls within the aims and scope of the journal. It is relevant in the context of the urgent need to approach Planetary Health, especially with health students.

Some particular suggestions/comments will be done here:

-      Abstract: reading the abstract and the scope of the manuscript, I miss any reference to animal health as they are also included in food systems and in other zoonoses transmitted in other forms rather than foodborne. One Health is included in Planetary Health and may not be neglected. If the authors want to reach only the vegetable part of food systems/agriculture, I believe Planetary Health is too ambitious for the title

-      Lines 53-54 – the authors should state at least the university/country of this Department

-      Lines 92 – 101 – too many lines for one only reference

-      Line 54 – LCME – first time write in full please

-      Line 104 – Figure 1 – what is “hi” cholesterol?

-      Line 131 – The authors should think they are not writing only to American readers, but for global (planetary!!) ones. So, whenever you refer concrete American projects, you should detail something more about them. This reviewer has no idea about what is “Farm Bill”.

-      Lines 181/182 – why did veterinarians were not included? It is a pity. The “animal” part is still missing and not only in the abstract.

-      Line 202 – ON – how did you assess all this “gain”? Maybe these are the project aims but not necessarily achieved? If you did assess, you should state here how did you do it.

-      Line 238 – One Health and Planetary Health are two fashion areas, but they are two different concepts. Along the manuscript the authors seem to use one or another randomly and that should not happen.

Response

Thank you for your comments and suggestions. We have addressed specific comments below. We hope these have improved our project report.

-      Lines 53-54 – We have clarified this is a US department

-      Lines 92-101 – We have added references and better explained the points in this paragraph

-      Line 54 – We have defined LCME

-      Figure 1 – We have corrected “hi” to “high” cholesterol

-      Line 131 – We have defined the US Farm Bill and clarified the manuscript for an international audience

-      Lines 181-182 – We have added how our elective included animal-related content and specified where veterinary expertise was utilized.

-      Line 202 – Insights gained were assessed through institutional course feedback in the form of a debriefing session. An IRB approval was not sought and thus we cannot report specific student quotes and comments. To address your concerns, we have added detail and examples of specific outcomes from the perspective of the students. Additionally, student participation in this manuscript reflects the positive outcomes of this elective beyond project aims.

-      Line 238 – We have clarified how our elective included animal content where veterinary expertise was utilized. We have removed mentions of One Health to be consistent with our focus on Planetary Health.